# Exponentially Weighted Imitation Learning for Batched Historical Data

**Qing Wang[1]    Jiechao Xiong[1]    Lei Han[1]    Peng Sun[1]    Han Liu[12]    Tong Zhang[1]**
[1]Tencent AI Lab    [2]Northwestern University
{drwang, jcxiong, lxhan, pythonsun}@tencent.com
hanliu@northwestern.edu, tongzhang@tongzhang-ml.org

## Abstract

We consider deep policy learning with only batched historical trajectories. The main challenge of this problem is that the learner no longer has a simulator or "environment oracle" as in most reinforcement learning settings. To solve this problem, we propose a monotonic advantage reweighted imitation learning strategy that is applicable to problems with complex nonlinear function approximation and works well with hybrid (discrete and continuous) action space. The method does not rely on the knowledge of the behavior policy, thus can be used to learn from data generated by an unknown policy. Under mild conditions, our algorithm, though surprisingly simple, has a policy improvement bound and outperforms most competing methods empirically. Thorough numerical results are also provided to demonstrate the efficacy of the proposed methodology.

## 1  Introduction

In this article, we consider the problem of learning a deep policy with batched historical trajectories. This problem is important and challenging. As in many real-world tasks, we usually have numerous historical data generated by different policies, but is lack of a perfect simulator of the environment. In this case, we want to learn a good policy from these data, to make decisions in a complex environment with possibly continuous state space and hybrid action space of discrete and continuous parts.

Several existing fields of research concern the problem of policy learning from batched data. In particular, imitation learning (IL) aims to find a policy whose performance is close to that of the data-generating policy [Abbeel and Ng, 2004]. On the other hand, off-policy reinforcement learning (RL) concerns the problem of learning a good (or possibly better) policy with data collected from a *behavior* policy [Sutton and Barto, 1998]. However, to the best of our knowledge, previous methods do not have satisfiable performance or are not directly applicable in a complex environment as ours with continuous state and hybrid action space.

In this work, we propose a novel yet simple method, to imitate a better policy by monotonic advantage reweighting. From theoretical analysis and empirical results, we find the proposed method has several advantages that

- From theoretical analysis, we show that the algorithm as proposed has policy improvement lower bound under mild condition.

- Empirically, the proposed method works well with function approximation and hybrid action space, which is crucial for the success of deep RL in practical problems.

- For off-policy learning, the method does not rely on the knowledge of action probability of the behavior policy, thus can be used to learn from data generated by an unknown policy, and is robust when current policy is deviated from the behavior policy.

In our real-world problem of a complex MOBA game, the proposed method has been successfully applied on human replay data, which validates the effectiveness of the method.

The article is organized as follows: We firstly state some preliminaries (Sec. 2) and related works (Sec. 3). Then we present our main method of imitating a better policy (Sec. 4), with theoretical analysis (Sec. 5) and empirical experiments (Sec. 6). Finally we conclude our discussion (Sec. 7).

## 2 Preliminaries

Consider a Markov decision process (MDP) with infinite-horizon, denoted by $\mathcal{M} = (\mathcal{S}, \mathcal{A}, P, r, d_0, \gamma)$, where $\mathcal{S}$ is the state space, $\mathcal{A}$ is the action space, $P$ is the transition probability defined on $\mathcal{S} \times \mathcal{A} \times \mathcal{S} \to [0,1]$, $r$ is the reward function $\mathcal{S} \times \mathcal{A} \to \mathbb{R}$, $d_0$ is the distribution of initial state $s_0$, and $\gamma \in (0,1)$ is the discount factor. A trajectory $\tau$ is a sequence of triplets of state, action and reward, i.e., $\tau = \{(s_t, a_t, r_t)\}_{t=1,\dots,T}$, where $T$ is the terminal step number. A stochastic policy denoted by $\pi$ is defined as $\mathcal{S} \times \mathcal{A} \to [0,1]$. We use the following standard notation of state-value $V^\pi(s_t)$, action-value $Q^\pi(s_t, a_t)$ and advantage $A^\pi(s_t, a_t)$, defined as $V^\pi(s_t) = \mathbb{E}_{\pi|s_t} \sum_{l=0}^\infty \gamma^l r(s_{t+l}, a_{t+l})$, $Q^\pi(s_t, a_t) = \mathbb{E}_{\pi|s_t, a_t} \sum_{l=0}^\infty \gamma^l r(s_{t+l}, a_{t+l})$, and $A^\pi(s_t, a_t) = Q^\pi(s_t, a_t) - V^\pi(s_t)$, where $\mathbb{E}_{\pi|s_t}$ means $a_l \sim \pi(a|s_l)$, $s_{l+1} \sim P(s_{l+1}|s_l, a_l)$, $\forall l \geq t$, and $\mathbb{E}_{\pi|s_t, a_t}$ means $s_{l+1} \sim P(s_{l+1}|s_l, a_l)$, $a_{l+1} \sim \pi(a|s_{l+1})$, $\forall l \geq t$. As the state space $\mathcal{S}$ may be prohibitively large, we approximate the policy and state-value with parameterized forms as $\pi_\theta(s, a)$ and $V_\theta^\pi(s)$ with parameter $\theta \in \Theta$. We denote the original policy space as $\Pi = \{\pi | \pi(s, a) \in [0,1], \sum_{a \in \mathcal{A}} \pi(s, a) = 1, \forall s \in \mathcal{S}, a \in \mathcal{A}\}$ and parametrized policy space as $\Pi_\Theta = \{\pi_\theta | \theta \in \Theta\}$.

To measure the similarity between two policies $\pi$ and $\pi'$, we consider the Kullback–Leibler divergence and total variance (TV) distance defined as

$$
\begin{aligned}
D_{\mathrm{KL}}^d(\pi' || \pi) &= \sum_s d(s) \sum_a \pi'(a|s) \log \frac{\pi'(a|s)}{\pi(a|s)} \\
D_{\mathrm{TV}}^d(\pi', \pi) &= (1/2) \sum_s d(s) \sum_a |\pi'(a|s) - \pi(a|s)|
\end{aligned}
$$

where $d(s)$ is a probability distribution of states. The performance of a policy $\pi$ is measured by its expected discounted reward:

$$
\eta(\pi) = \mathbb{E}_{d_0, \pi} \sum_{t=0}^\infty \gamma^t r(s_t, a_t)
$$

where $\mathbb{E}_{d_0, \pi}$ means $s_0 \sim d_0$, $a_t \sim \pi(a_t|s_t)$, and $s_{t+1} \sim P(s_{t+1}|s_t, a_t)$. We omit the subscript $d_0$ when there is no ambiguity. In [Kakade and Langford, 2002], a useful equation has been proved that

$$
\eta(\pi') - \eta(\pi) = \frac{1}{1-\gamma} \sum_s d_{\pi'}(s) \sum_a \pi'(a|s) A^\pi(s, a)
$$

where $d_\pi$ is the discounted visiting frequencies defined as $d_\pi(s) = (1-\gamma)\mathbb{E}_{d_0, \pi} \sum_{t=0}^\infty \gamma^t \mathbf{1}(s_t = s)$ and $\mathbf{1}(\cdot)$ is an indicator function. In addition, define $L^{d,\pi}(\pi')$ as

$$
L^{d,\pi}(\pi') = \frac{1}{1-\gamma} \sum_s d(s) \sum_a \pi'(a|s) A^\pi(s, a)
$$

then from [Schulman et al., 2015, Theorem 1], the difference of $\eta(\pi')$ and $\eta(\pi)$ can be approximated by $L^{d_\pi, \pi}(\pi')$, where the approximation error is bounded by total variance $D_{\mathrm{TV}}^{d_\pi}(\pi', \pi)$, which can be further bounded by $D_{\mathrm{KL}}^{d_\pi}(\pi' || \pi)$ or $D_{\mathrm{KL}}^{d_\pi}(\pi || \pi')$.

In the following sections, we mainly focus on maximizing $L^{d_\pi, \pi}(\pi_\theta)$ as a proxy for optimizing policy performance $\eta(\pi_\theta)$, for $\pi_\theta \in \Pi_\Theta$.

## 3 Related Work

Off-policy learning [Sutton and Barto, 1998] is a broad region of research. For policy improvement method with performance guarantee, *conservative policy iteration* [Kakade and Langford, 2002] or

*safe policy iteration* [Pirotta et al., 2013] has long been an interesting topic in the literature. The term "safety" or "conservative" usually means the algorithm described is guaranteed to produce a series of monotonic improved policies. Exact or high-probability bounds of policy improvement are often provided in these previous works [Thomas and Brunskill, 2016, Jiang and Li, 2016, Thomas et al., 2015, Ghavamzadeh et al., 2016]. We refer readers to [García and Fernández, 2015] for a comprehensive survey of safe RL. However, to the best of our knowledge, these prior methods cannot be directly applied in our problem of learning in a complex game environment with large scale replay data, as they either need full-knowledge of the MDP or consider tabular case mainly for finite states and discrete actions, with prohibitive computational complexity.

Constrained policy optimization problems in the parameter space are considered in previous works [Schulman et al., 2015, Peters et al., 2010]. In [Peters et al., 2010], they constrain the policy on the distribution of $p^\pi(s, a) = \mu^\pi(s)\pi(a|s)$, while in [Schulman et al., 2015], the constraint is on $\pi(a|s)$, with fixed state-wise weight $d(s)$. Also, in [Schulman et al., 2015], the authors have considered $D_{\mathrm{KL}}^{d_\pi}(\pi||\pi_\theta)$ as a policy divergence constraint, while in [Peters et al., 2010] the authors considered $D_{\mathrm{KL}}(\mu^\pi\pi||q)$. The connection with our proposed method is elaborated in Appendix B.1. A closely related work is [Abdolmaleki et al., 2018] which present the exponential advantage weighting in an EM perspective. Independently, we further generalize to monotonic advantage re-weighting and also derive a lower bound for imitation learning.

Besides off-policy policy iteration algorithm, value iteration algorithm can also be used in off-policy settings. For deep reinforcement learning, DQN [Mnih et al., 2013], DQfD [Hester et al., 2018] works primarily with discrete actions, while DDPG [Lillicrap et al., 2016] works well with continuous actions. For hybrid action space, there are also works combining the idea of DQN and DDPG [Hausknecht and Stone, 2016]. In our preliminary experiments, we found value iteration method failed to converge for the tasks in the HFO environment. It seems that the discrepancy between behavior policy and the target policy ($\arg\max$ policy in DQN) should be properly restrained, which we think worth further research and investigation.

Also, there are existing related methods in the field of imitation learning. For example, when expert data is available, we can learn a policy directly by predicting the expert action [Bain and Sommut, 1999, Ross et al., 2011]. Another related idea is to imitate an MCTS policy [Guo et al., 2014, Silver et al., 2016]. In the work of [Silver et al., 2016], the authors propose to use Monte-Carlo Tree Search (MCTS) to form a new policy $\tilde{\pi} = \mathrm{MCTS}(\pi)$ where $\pi$ is the base policy of network, then imitate the better policy $\tilde{\pi}$ by minimizing $D_{\mathrm{KL}}(\tilde{\pi}||\pi_\theta)$. Also in [Guo et al., 2014], the authors use UCT as a policy improvement operator and generate data from $\tilde{\pi} = \mathrm{UCT}(\pi)$, then perform regression or classification with the dataset, which can be seen as approximating the policy under normal distribution or multinomial distribution parametrization.

## 4    Monotonic Advantage Re-Weighted Imitation Learning (MARWIL)

To learn a policy from data, the most straight forward way is imitation learning (behavior cloning). Suppose we have state-action pairs $(s_t, a_t)$ in the data generated by a behavior policy $\pi$, then we can minimize the KL divergence between $\pi$ and $\pi_\theta$. To be specific, we would like to minimize

$$D_{\mathrm{KL}}^d(\pi||\pi_\theta) = -\mathbb{E}_{s\sim d(s), a\sim\pi(a|s)}(\log\pi_\theta(a|s) - \log\pi(a|s)) \tag{1}$$

under some state distribution $d(s)$. However, this method makes no distinction between "good" and "bad" actions. The learned $\pi_\theta$ simply imitates all the actions generated by $\pi$. Actually, if we also have reward $r_t$ in the data, we can know the consequence of taking action $a_t$, by looking at future state $s_{t+1}$ and reward $r_t$. Suppose we have estimation of the advantage of action $a_t$ as $\widehat{A}^\pi(s_t, a_t)$, we can put higher sample weight on the actions with higher advantage, thus imitating good actions more often. Inspired by this idea, we propose a monotonic advantage reweighted imitation learning method (Algorithm 1) which maximizes

$$\mathbb{E}_{s\sim d_\pi(s), a\sim\pi(a|s)} \exp(\beta\widehat{A}^\pi(s, a)) \log\pi_\theta(a|s) \tag{2}$$

where $\beta$ is a hyper-parameter. When $\beta = 0$ the algorithm degenerates to ordinary imitation learning. Ideally we would like to estimate the advantage function $A(s_t, a_t) = \mathbb{E}_{\pi|s_t, a_t}(R_t - V^\pi(s_t))$ using cumulated discounted future reward $R_t = \sum_{l=t}^{T} \gamma^{l-t} r_l$. For example, one possible solution is to use a neural network to estimate $A(s_t, a_t)$, by minimizing $\mathbb{E}_{\pi|s_t, a_t}(A_\theta(s_t, a_t) - (R_t - V_\theta(s_t)))^2$

---

**Algorithm 1** Monotonic Advantage Re-Weighted Imitation Learning (MARWIL)

---

**Input:** Historical data $\mathcal{D}$ generated by $\pi$, hyper-parameter $\beta$.

For each trajectory $\tau$ in $\mathcal{D}$, estimate advantages $\widehat{A}^\pi(s_t, a_t)$ for time $t = 1, \cdots, T$.

Maximize $\mathbb{E}_{(s_t, a_t) \in \mathcal{D}} \exp(\beta \widehat{A}^\pi(s_t, a_t)) \log \pi_\theta(a_t|s_t)$ with respect to $\theta$.

---

for $R_t$ computed from different trajectories, where $V_\theta(s_t)$ is also estimated with a neural network respectively. In practice we find that good results can be achieved by simply using a *single path* estimation as $\widehat{A}(s_t, a_t) = (R_t - V_\theta(s_t))/c$, where we normalize the advantage by its average norm $c$[1] in order to make the scale of $\beta$ stable across different environments. We use this method in our experiments as it greatly simplifies the computation.

Although the algorithm has a very simple formulation, it has many strengths as

1. Under mild conditions, we show that the proposed algorithm has policy improvement bound by theoretical analysis. Specifically, the policy $\tilde{\pi}$ is uniformly as good as, or better than the behavior policy $\pi$.

2. The method works well with function approximation as a complex neural network, as suggested by theoretical analysis and validated empirically. The method is naturally compatible with hybrid action of discrete and continuous parts, which is common in practical problems.

3. In contrast to most off-policy methods, the algorithm does not rely on importance sampling with the value of $\pi(a_t|s_t)$ – the action probability of the behavior policy, thus can be used to learn from an unknown policy, and is also robust when current policy is deviated from the behavior policy. We validate this with several empirical experiments.

In Section 5 we give a proposition of policy improvement by theoretical analysis. And in Section 6 we give experimental results of the proposed algorithm in off-policy settings.

## 5 Theoretical Analysis

In this section, we firstly show that in the ideal case Algorithm 1 is equivalent to imitating a new policy $\tilde{\pi}$. Then we show that the policy $\tilde{\pi}$ is indeed uniformly better than $\pi$. Thus Algorithm 1 can also be regarded as *imitating a better policy* (IBP). For function approximation, we also provide a policy improvement lower bound under mild conditions.

### 5.1 Equivalence to Imitating a New Policy

In this subsection, we show that in the ideal case when we know the advantage $A^\pi(s_t, a_t)$, Algorithm 1 is equivalent to minimizing KL divergence between $\pi_\theta$ and a hypothetic $\tilde{\pi}$. Consider the problem

$$\tilde{\pi} = \arg\max_{\pi' \in \Pi} ((1 - \gamma)\beta L^{d_\pi, \pi}(\pi') - D_{\text{KL}}^{d_\pi}(\pi'||\pi)) \tag{3}$$

which has an analytical solution in the policy space $\Pi$ [Azar et al., 2012, Appendix A, Proposition 1]

$$\tilde{\pi}(a|s) = \pi(a|s) \exp(\beta A^\pi(s, a) + C(s)) \tag{4}$$

where $C(s)$ is a normalizing factor to ensure that $\sum_{a \in \mathcal{A}} \tilde{\pi}(a|s) = 1$ for each state $s$. Then

$$\arg\min_\theta D_{\text{KL}}^d(\tilde{\pi}||\pi_\theta) = \arg\max_\theta \sum_s d(s) \sum_a \tilde{\pi}(a|s) \log \pi_\theta(a|s)$$

$$= \arg\max_\theta \sum_s d(s) \exp(C(s)) \sum_a \pi(a|s) \exp(\beta A^\pi(s, a)) \log \pi_\theta(a|s) \tag{5}$$

Thus Algorithm 1 is equivalent to minimizing $D_{\text{KL}}^d(\tilde{\pi}||\pi_\theta)$ for $d(s) \propto d_\pi(s) \exp(-C(s))$. [2]

## 5.2 Monotonic Advantage Reweighting

In subsection 5.1, we have shown that the $\tilde{\pi}$ defined in 4 is the analytical solution to the problem 3. In this section, we further show that $\tilde{\pi}$ is indeed uniformly as good as, or better than $\pi$. To be rigorous, a policy $\pi'$ is considered *uniformly as good as, or better* than $\pi$, if $\forall s \in \mathcal{S}$, we have $V^{\pi'}(s) \geq V^{\pi}(s)$. In Proposition 1, we give a family of $\tilde{\pi}$ which are uniformly as good as, or better than $\pi$. To be specific, we have

**Proposition 1.** *Suppose two policies $\pi$ and $\tilde{\pi}$ satisfy*

$$g(\tilde{\pi}(a|s)) = g(\pi(a|s)) + h(s, A^{\pi}(s, a)) \tag{6}$$

*where $g(\cdot)$ is a monotonically increasing function, and $h(s, \cdot)$ is monotonically increasing for any fixed $s$. Then we have*

$$V^{\tilde{\pi}}(s) \geq V^{\pi}(s), \ \forall s \in S. \tag{7}$$

*that is, $\tilde{\pi}$ is uniformly as good as or better than $\pi$.*

The idea behind this proposition is simple. The condition (6) requires that the policy $\tilde{\pi}$ has positive advantages for the actions where $\tilde{\pi}(a|s) \geq \pi(a|s)$. Then it follows directly from the well-known *policy improvement theorem* as stated in [Sutton and Barto, 1998, Equation 4.8]. A short proof is provided in Appendix A.1 for completeness.

When $g(\cdot)$ and $h(s, \cdot)$ in (6) are chosen as $g(\pi) = \log(\pi)$ and $h(s, A^{\pi}(s, a)) = \beta A^{\pi}(s, a) + C(s)$, then we recover the formula in 4. By Proposition (1) we have shown that $\tilde{\pi}$ defined in 4 is as good as, or better than policy $\pi$.

We note that there are other choice of $g(\cdot)$ and $h(s, \cdot)$ as well. For example we can choose $g(\pi) = \log(\pi)$ and $h(s, A^{\pi}(s, a)) = \log((\beta A^{\pi}(s, a))_+ + \epsilon) + C(s)$, where $(\cdot)_+$ is a positive truncation, $\epsilon$ is a small positive number, and $C(s)$ is a normalizing factor to ensure $\sum_{a \in \mathcal{A}} \tilde{\pi}(s, a) = 1$. In this case, we can minimize $D_{KL}^d(\tilde{\pi}||\pi_\theta) = \sum_s d(s) \exp(C(s)) \sum_a \pi(a|s)((\beta A^{\pi}(s, a))_+ + \epsilon) \log \pi_\theta(a|s) + C$.

## 5.3 Lower bound under Approximation

For practical usage, we usually seek a parametric approximation of $\tilde{\pi}$. The following proposition gives a lower bound of policy improvement for the parametric policy $\pi_\theta$.

**Proposition 2.** *Suppose we use parametric policy $\pi_\theta$ to approximate the improved policy $\tilde{\pi}$ defined in Formula 3, we have the following lower bound on the policy improvement*

$$\eta(\pi_\theta) - \eta(\pi) \geq -\frac{\sqrt{2}}{1-\gamma} \delta_1^{\frac{1}{2}} M^{\pi_\theta} + \frac{1}{(1-\gamma)\beta} \delta_2 - \frac{\sqrt{2}\gamma\epsilon_{\tilde{\pi}}}{(1-\gamma)^2} \delta_2^{\frac{1}{2}} \tag{8}$$

*where $\delta_1 = \min(D_{KL}^{d_{\tilde{\pi}}}(\pi_\theta||\tilde{\pi}), D_{KL}^{d_{\tilde{\pi}}}(\tilde{\pi}||\pi_\theta))$, $\delta_2 = D_{KL}^{d_\pi}(\tilde{\pi}||\pi)$, $\epsilon_{\pi}^{\pi'} = \max_s |\mathbb{E}_{a \sim \pi'} A^{\pi}(s, a)|$, and $M^{\pi} = \max_{s,a} |A^{\pi}(s, a)| \leq \max_{s,a} |r(s, a)|/(1 - \gamma)$.*

A short proof can be found in Appendix A.2. Note that we would like to approximate $\tilde{\pi}$ under state distribution $d_{\tilde{\pi}}$ in theory. However in practice we use a heuristic approximation to sample data from trajectories generated by the base policy $\pi$ as in Algorithm 1, which is equivalent to imitating $\tilde{\pi}$ under a slightly different state distribution $d$ as discussed in Sec.5.1.

# 6 Experimental Results

In this section, we provide empirical evidence that the algorithm is well suited for off-policy RL tasks, as it does not need to know the probability of the behavior policy, thus is robust when learning from replays from an unknown policy. We evaluate the proposed algorithm with HFO environment under different settings (Sec. 6.1). Furthermore, we also provide two other environments (TORCS and mobile MOBA game) to evaluate the algorithm in learning from replay data (Sec. 6.2, 6.3).

Denote the behavior policy as $\pi$, the desired parametrized policy as $\pi_\theta$, the policy loss $L_p$ for the policy iteration algorithms considered are listed as following: ($C$ is a $\theta$-independent constant)

- **(IL)** Imitation learning, minimizing $D_{KL}^{d_\pi}(\pi||\pi_\theta)$.

$$L_p = D_{KL}^{d_\pi}(\pi||\pi_\theta) = -\mathbb{E}_{s \sim d_\pi(s), a \sim \pi(a|s)} \log \pi_\theta(a|s) + C \tag{9}$$

- (**PG**) Policy gradient with baseline and $D_{\text{KL}}^{d_\pi}(\pi||\pi_\theta)$ regularization.

$$L_p = -\mathbb{E}_{s\sim d_\pi(s), a\sim\pi(a|s)}(\beta A^\pi(s,a) + 1)\log\pi_\theta(a|s) + C \tag{10}$$

- (**PGIS**) Policy gradient with baseline and $D_{\text{KL}}^{d_\pi}(\pi||\pi_\theta)$ regularization, with off-policy correction by importance sampling (IS), as in TRPO [Schulman et al., 2015] and CPO [Achiam et al., 2017]. Here we simply use penalized gradient algorithm to optimize the objective, instead of using delegated optimization method as in [Schulman et al., 2015].

$$\begin{aligned} L_p &= D_{\text{KL}}^{d_\pi}(\pi||\pi_\theta) - (1-\gamma)\beta L^{d_\pi,\pi}(\pi_\theta) \\ &= -\mathbb{E}_{s\sim d_\pi(s), a\sim\pi(a|s)}\left(\frac{\pi_\theta(a|s)}{\pi(a|s)}\beta A^\pi(s,a) + \log\pi_\theta(s,a)\right) + C \end{aligned} \tag{11}$$

- (**MARWIL**) Minimizing $D_{\text{KL}}^d(\tilde{\pi}||\pi_\theta)$ as in (5) and Algorithm 1.

$$L_p = D_{\text{KL}}^d(\tilde{\pi}||\pi_\theta) = -\mathbb{E}_{s\sim d_\pi(s), a\sim\pi(a|s)}\log(\pi_\theta(a|s))\exp(\beta A^\pi(s,a)) + C \tag{12}$$

Note that IL simply imitates all the actions in the data, while PG needs the on-policy assumption to be a reasonable algorithm. Both PGIS and MARWIL are derived under off-policy setting. However, the importance ratio $\pi_\theta/\pi$ used to correct off-policy bias for PG usually has large variance and may cause severe problems when $\pi_\theta$ is deviated far away from $\pi$ [Sutton and Barto, 1998]. Several methods are proposed to alleviate this problem [Schulman et al., 2017, Munos et al., 2016, Precup et al., 2000]. On the other hand, we note that the algorithm MARWIL is naturally off-policy, instead of relying on the importance sampling ratio $\pi_\theta/\pi$ to do off-policy correction. We expect the proposed algorithm to work better when learning from a possibly unknown behavior policy.

## 6.1 Experiments with Half Field Offense (HFO)

To compare the aforementioned algorithms, we employ Half Field Offense (HFO) as our primary experiment environment. HFO is an abstraction of the full RoboCup 2D game, where an agent plays soccer in a half field. The HFO environment has continuous state space and hybrid (discrete and continuous) action space, which is similar to our task in a MOBA game (Sec. 6.3). In this simplified environment, we validate the effectiveness and efficiency of the proposed learning method.

### 6.1.1 Environment Settings

Like in [Hausknecht and Stone, 2016], we let the agent try to goal without a goalkeeper. We follow [Hausknecht and Stone, 2016] for the settings, as is briefed below.

The observation is a 59-d feature vector, encoding the relative position of several critical objects such as the ball, the goal and other landmarks (See [Hausknecht, 2017]). In our experiments, we use a hybrid action space of discrete actions and continuous actions. 3 types of actions are considered in our setting, which correspond to {"Dash", "Turn", "Kick"}. For each type $k$ of action, we require the policy to output a parameter $x_k \in \mathbb{R}^2$. For the action "Dash" and "Kick", the parameter $x_k$ is interpreted as $(r\cos\alpha, r\sin\alpha)$, with $r$ truncated to 1 when exceeding. Then $\alpha \in [0, 2\pi]$ is interpreted as the relative direction of that action, while $r \in [0,1]$ is interpreted as the power/force of that action. For the action "Turn", the parameter $x_k$ is firstly normalized to $(\cos\alpha, \sin\alpha)$ and then $\theta$ is interpreted as the relative degree of turning. The reward is hand-crafted, written as:

$$r_t = d_t(b,a) - d_{t+1}(b,a) + \mathbb{I}_{t+1}^{kick} + 3(d_t(b,g) - d_{t+1}(b,g)) + 5\mathbb{I}_{t+1}^{goal},$$

where $d_t(b,a)$ (or $d_t(b,g)$) is the distance between the ball and the agent (or the center of goal). $\mathbb{I}_t^{kick} = 1$ if the agent is close enough to kick the ball. $\mathbb{I}_t^{goal} = 1$ if a successful goal happens. We leverage *Winning Rate* $= \frac{N_{\text{G}}}{N_{\text{G}}+N_{\text{F}}}$ to evaluate the final performance, where $N_{\text{G}}$ is the number of goals (G) achieved, $N_{\text{F}}$ is the number of failures (F), due to either out-of-time (the agent does not kick the ball in 100 frames or does not goal in 500 frames) or out-of-bound (the ball is out of the half field).

When learning from data, the historical experience is generated with a mixture of a perfect (100% winning rate) policy $\pi_{\text{perfect}}$ and a random policy $\pi_{\text{random}}$. For the continuous part of the action, a Gaussian distribution of $\sigma = 0.2$ or $0.4$ is added to the model output, respectively. The mixture

---
**Algorithm 2** Stochastic Gradient Algorithm for MARWIL
---
**Input:** Policy loss $L_p$ being one of 9 to 12. base policy $\pi$, parameter $m, c_v$.
Randomly initialize $\pi_\theta$. Empty replay memory $D$.
Fill $D$ with trajectories from $\pi$ and calculate $R_t$ for each $(s_t, a_t)$ in $D$.
**for** $i = 1$ **to** $N$ **do**
    Sample a batch $B = \{(s_k, a_k, R_k)\}_m$ from D.
    Compute mini-batch gradient $\nabla_\theta \widehat{L}_p, \nabla_\theta \widehat{L}_v$ of $B$.
    Update $\theta$: $-\Delta\theta \propto \nabla_\theta \widehat{L}_p + c_v \nabla_\theta \widehat{L}_v$
**end for**
---

Table 1: Performance of PG and MARWIL in TORCS, where $\beta = 0$ is the case of IL. For consistent performance, $\beta$ should be inversely proportional to the scale of (normalized) $A^\pi$. Different $\beta$ are tested in the experiments. The performance is evaluated on the sum of rewards per episode.

| $\beta$ | 0.0 | 0.25 | 0.5 | 0.75 | 1.0 |
|---|---|---|---|---|---|
| PG | 2710 | 6396 | 6735 | 6758 | 7152 |
| MARWIL | (2710) | 5583 | 6832 | 7670 | 9492 |

coefficient $\epsilon$ is used to adjust the proportion of "good" actions and "bad" actions. To be specific, for each step, the action is taken as

$$a_t \sim \begin{cases} \pi_{\text{perfect}}(\cdot|s_t) + N(0, \sigma) & \text{w.p. } \epsilon \\ \pi_{\text{random}}(\cdot|s_t) + N(0, \sigma) & \text{w.p. } 1 - \epsilon \end{cases} \tag{13}$$

The parameter $\epsilon$ is adjusted from $0.1$ to $0.5$. Smaller $\epsilon$ means greater noise, in which case it is harder for the algorithms to find a good policy from the noisy data.

### 6.1.2 Algorithm Setting

For the HFO game, we model the 3 discrete actions with multinomial probabilities and the 2 continuous parameters for each action with normal distributions of known $\sigma = 0.2$ but unknown $\mu$. Parameters for different types of action are modeled separately. In total we have 3 output nodes for discrete action probabilities and 6 output nodes for continuous action parameters, in the form of

$$\pi_\theta((k, x_k)|s) = p_\theta(k|s)N(x_k|\mu_{\theta,k}, \sigma), \quad k \in \{1, 2, 3\}, x_k \in \mathbb{R}^2$$

where $p_\theta(\cdot|s)$ is computed as a soft-max for discrete actions and $N(\cdot|\mu_\theta, \sigma)$ is the probability density function of Gaussian distribution.

When learning from data, the base policy (13) is used to generate trajectories into a replay memory $D$, and the policy network is updated by different algorithms, respectively. We denote the policy loss objective as $L_p$, being one of the formula (9) (10) (11) (12). Then we optimize the policy loss $L_p$ and the value loss $L_v$ simultaneously, with a mixture coefficient $c_v$ as a hyper-parameter (by default $c_v = 1$). The value loss $L_v$ is defined as $L_v = \mathbb{E}_{d,\pi}(R_t - V_\theta(s_t))^2$. A stochastic gradient algorithm is given in Algorithm 2. Each experiment is repeated 3 times and the average of scores is reported in Figure 1. Additional details of the algorithm settings are given in Appendix B.2.

We note that the explicit value $\pi(a_t|s_t)$ is crucial for the correction used by most off-policy policy iteration methods [Sutton and Barto, 1998], including [Munos et al., 2016, Wang et al., 2016, Schulman et al., 2017, Wu et al., 2017] and many other works [Geist and Scherrer, 2014]. Here for a comparable experiment between policy gradient method and our proposed method, we consider a simple off-policy correction by importance sampling as in (11). We test the performance of the proposed method and previous works under different settings in Figure 1. We can see that the proposed MARWIL achieves consistently better performance than other methods.[3]

### 6.2 Experiments with TORCS

We also evaluate the imitation learning and the proposed method within the TORCS [Wymann et al., 2014] environment. In the TORCS environment, the observation is the raw screen with image size of

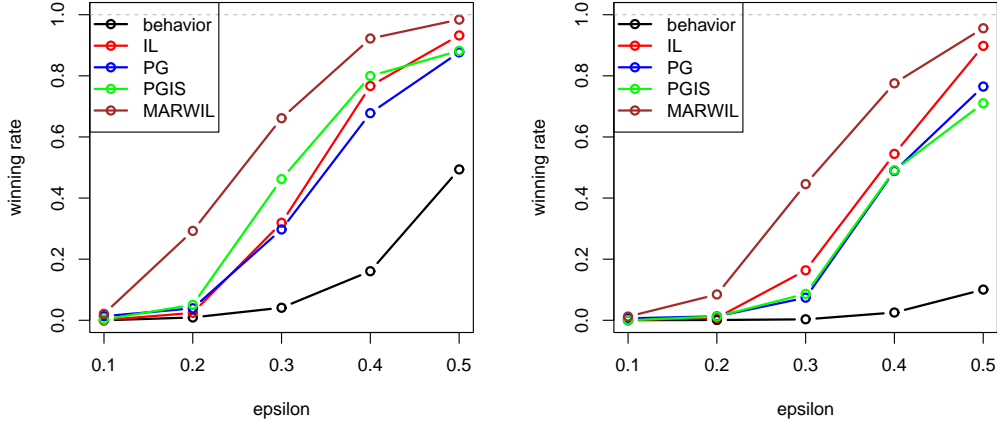

Figure 1: **Left:** Learning from data with additional noise $\sigma = 0.2$. **Right:** Learning from data with additional noise $\sigma = 0.4$. The data is generated with a mixture of a perfect (100% winning rate) policy $\pi_{\text{perfect}}$ and a random policy $\pi_{\text{random}}$. For the continuous part of the action, a Gaussian noise of $\sigma = 0.2$ (left) or $0.4$ (right) is added to the model output, respectively. The mixture coefficient $\epsilon$ is used to adjust the proportion of "good" actions and "bad" actions. Smaller $\epsilon$ means less "good" actions and harder problem. The performance of the behavior policy is plotted in black. We see that the performance of IL is stable, while PG and PGIS may be affected by the increasing noise in the data. In all settings we see that the proposed algorithm MARWIL performs best in this task.

$64 \times 64 \times 3$, the action is a scalar indicating the steering angle in $[-\pi, \pi]$, and the reward $r_t$ is the momentary speed. When the car crushes, a $-1$ reward is received and the game terminates.

For the TORCS environment, a simple rule is leveraged to keep the car running and to prevent it from crushing. Therefore, we can use the rule as the optimal policy to generate expert trajectories. In addition, we generate noisy trajectories with random actions to intentionally confuse the learning algorithms, and see whether the proposed method can learn a better policy from the data generated by the deteriorated policy. We make the training data by generating 10 matches with the optimal policy and another 10 matches with the random actions.

We train the imitation learning and the proposed method for 5 epochs to compare their performance. Table 1 shows the test scores when varying the parameter $\beta$. From the results, we see that our proposed algorithm is effective at learning a better policy from these noisy trajectories.

### 6.3 Experiments with King of Glory

We also evaluate the proposed algorithm with *King of Glory* – a mobile MOBA (Multi-player Online Battle Arena) game popular in China. In the experiments, we collect human replay files in the size of millions, equaling to tens of billions time steps in total. Evaluation is performed in the "solo" game mode, where an agent fights against another AI in the opposite side. A DNN based function approximator is adopted. In a proprietary test, we find that our AI agent, trained with the proposed method, can reach the level of an experienced human player in a solo game. Additional details of the algorithm settings for King of Glory is given in Appendix B.3.

## 7 Conclusion

In this article, we present an off-policy learning algorithm that can form a better policy from trajectories generated by a possibly unknown policy. When learning from replay data, the proposed algorithm does not require the bahavior probability $\pi$ over the actions, which is usually missing in human generated data, and also works well with function approximation and hybrid action space. The algorithm is preferable in real-world application, including playing video games. Experimental results over several real world datasets validate the effectiveness of the proposed algorithm. We note that the proposed MARWIL algorithm can also work as a full reinforcement learning method, when applied iteratively on self-generated replay data. Due to the space limitation, a thorough study of our method for full reinforcement learning is left to a future work.

**Acknowledgement** We are grateful for the anonymous reviewers for their detailed and helpful comments on this work. We also thank our colleagues in the project of King of Glory AI, particularly Haobo Fu and Tengfei Shi, for their assistance on the game environment and parsing replay data.

## Footnotes

[1] In our experiments, the average norm of advantage is approximated with a moving average estimation, by $c^2 \leftarrow c^2 + 10^{-8}((R_t - V_\theta(s_t))^2 - c^2)$.

[2] In the implementation of the algorithm, we omit the step discount in $d_\pi$, i.e., using $d'_\pi(s) = \mathbb{E}_{d_0, \pi} \sum_{t=0}^T \mathbf{1}(s_t = s)$ where $T$ is the terminal step. Sampling from $d_\pi(s)$ is possible, but usually leads to inferior performance according to our preliminary experiments.

[3]We note that the gap between behavior policy and IL is partly due to the approximation we used. As we have continuous action space, we use a gaussian model with fixed $\sigma$, thus the variance of learned policy may be lower than that of the behavior policy. A fair comparison should be made among IL, PG, PGIS, and MARWIL.

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
