[Supplementary Material]

# A Proofs

In this section, we provide proofs for the propositions appeared in the main text.

## A.1 Proof to Proposition 1

*Proof.* For a fixed $s$, consider $A_1 = \{a \in \mathcal{A} \mid \tilde{\pi}(a|s) \geq \pi(a|s)\}$, $A_2 = \{a \in \mathcal{A} \mid \tilde{\pi}(a|s) < \pi(a|s)\}$. Since $g(\cdot)$ and $h(s, \cdot)$ is monotonically increasing, we have

$$h(s, A^\pi(s, a_1)) = g(\tilde{\pi}(a_1|s)) - g(\pi(a_1|s))$$
$$\geq g(\tilde{\pi}(a_2|s)) - g(\pi(a_2|s))$$
$$= h(s, A^\pi(s, a_2)), \ \forall a_1 \in A_1, a_2 \in A_2$$

which means that $\exists \, q(s) \in \mathbb{R}$ s.t.

$$Q^\pi(s, a_1) \geq q(s) \geq Q^\pi(s, a_2), \quad \forall a_1 \in A_1, a_2 \in A_2$$

Thus

$$\sum_a \tilde{\pi}(a|s)Q^\pi(s, a) - \sum_a \pi(a|s)Q^\pi(s, a)$$
$$= \sum_{a \in A_1} (\tilde{\pi}(a|s) - \pi(a|s))Q^\pi(s, a) + \sum_{a \in A_2} (\tilde{\pi}(a|s) - \pi(a|s))Q^\pi(s, a)$$
$$\geq \sum_{a \in A_1} (\tilde{\pi}(a|s) - \pi(a|s))q(s) + \sum_{a \in A_2} (\tilde{\pi}(a|s) - \pi(a|s))q(s)$$
$$= q(s) \sum_a \tilde{\pi}(a|s) - q(s) \sum_a \pi(a|s) = 0$$

Define

$$V_l(s) = \begin{cases} \mathbb{E}_{a \sim \tilde{\pi}(s)} \left( \mathbb{E}_{s', r|s, a}(r + \gamma V_{l-1}(s')) \right), & l \geq 1 \\ V^\pi(s), & l = 0 \end{cases}$$

that is, the value of state $s$ if we follow $\tilde{\pi}$ in the first $l$ steps, and then follow $\pi$ in subsequent steps. So we have just proved

$$V_1(s) \geq V_0(s), \quad \forall s \in S.$$

By induction, we assume that $V_l(s) \geq V_{l-1}(s), \ \forall s \in S$, then

$$V_{l+1}(s) = \mathbb{E}_{a \sim \tilde{\pi}} \left( \mathbb{E}_{s', r|s, a}(r + \gamma V_l(s')) \right)$$
$$V_l(s) = \mathbb{E}_{a \sim \tilde{\pi}} \left( \mathbb{E}_{s', r|s, a}(r + \gamma V_{l-1}(s')) \right)$$

we have $V_{l+1}(s) \geq V_l(s), \ \forall s \in S$. For finite horizon MDP and infinite horizon MDP with $\gamma < 1$, we have

$$V^{\tilde{\pi}}(s) \geq V^\pi(s), \quad \forall s \in S$$

$\square$

The proof is for discrete action space only. However it could be generalized to continuous actions and hybrid actions without much difficulties.

## A.2 Proof to Proposition 2

*Proof.* In [Kakade and Langford, 2002] a useful equation is proved that

$$\eta(\pi') - \eta(\pi) = L^{d_{\pi'}, \pi}(\pi') = \frac{1}{1 - \gamma} \sum_s d_{\pi'}(s) \sum_a \pi'(a|s)A^\pi(s, a) \tag{14}$$

From Corollary 1 in [Achiam et al., 2017], we have

$$\eta(\pi') - \eta(\pi) \geq L^{d_\pi, \pi}(\pi') - \frac{2\gamma \epsilon_\pi^{\pi'}}{(1 - \gamma)^2} D_{\mathrm{TV}}^{d_\pi}(\pi', \pi) \tag{15}$$

where $\epsilon_\pi^{\pi'} = \max_s |\mathbb{E}_{a \sim \pi'} A^\pi(s, a)|$. Similarly, we also have

$$|\eta(\pi') - \eta(\pi)| = \frac{1}{1 - \gamma} \sum_s d_{\pi'}(s) \left| \sum_a (\pi'(a|s) - \pi(a|s))A^\pi(s, a) \right| \tag{16}$$

$$\leq \frac{1}{1 - \gamma} \sum_s d_{\pi'}(s) \sum_a |\pi'(a|s) - \pi(a|s)| \, |A^\pi(s, a)| \tag{17}$$

$$\leq \frac{2}{1 - \gamma} D_{\mathrm{TV}}^{d_{\pi'}}(\pi', \pi)M^\pi \tag{18}$$

where $M^\pi = \max_{s,a} |A^\pi(s,a)| \leq \max_{s,a} |r(s,a)|/(1-\gamma)$. Define

$$\mathcal{L}(\pi') = (1-\gamma)\beta L^{d_\pi,\pi}(\pi') - D_{\text{KL}}^{d_\pi}(\pi'||\pi) \tag{19}$$

Then

$$\tilde{\pi} = \arg\max_{\pi' \in \Pi} \mathcal{L}(\pi') \tag{20}$$

and $\mathcal{L}(\tilde{\pi}) \geq \mathcal{L}(\pi) = 0$. Now consider

$$\eta(\pi_\theta) - \eta(\pi) = (\eta(\pi_\theta) - \eta(\tilde{\pi})) + (\eta(\tilde{\pi}) - \eta(\pi)) \tag{21}$$

$$\geq -\frac{2}{1-\gamma} D_{\text{TV}}^{d_{\tilde{\pi}}}(\tilde{\pi}, \pi_\theta) M^{\pi_\theta} + L^{d_\pi,\pi}(\tilde{\pi}) - \frac{2\gamma\epsilon_\pi^{\tilde{\pi}}}{(1-\gamma)^2} D_{\text{TV}}^{d_\pi}(\tilde{\pi}, \pi) \tag{22}$$

$$\geq -\frac{2}{1-\gamma} D_{\text{TV}}^{d_{\tilde{\pi}}}(\tilde{\pi}, \pi_\theta) M^{\pi_\theta} + \frac{1}{(1-\gamma)\beta} D_{\text{KL}}^{d_\pi}(\tilde{\pi}||\pi) - \frac{2\gamma\epsilon_\pi^{\tilde{\pi}}}{(1-\gamma)^2} D_{\text{TV}}^{d_\pi}(\tilde{\pi}, \pi) \tag{23}$$

From Pinsker's inequality [Csiszar and Körner, 2011], we have

$$D_{\text{TV}}^{d}(\pi', \pi) = \sum_s d(s) D_{\text{TV}}(\pi'(\cdot|s), \pi(\cdot|s)) \tag{24}$$

$$\leq \sum_s d(s) \sqrt{\frac{1}{2} D_{\text{KL}}(\pi'(\cdot|s)||\pi(\cdot|s))} \leq \sqrt{\frac{1}{2} D_{\text{KL}}^d(\pi'||\pi)} \tag{25}$$

where the last inequality comes from Jensen's inequality. Denote $\delta_1 = \min(D_{\text{KL}}^{d_{\tilde{\pi}}}(\pi_\theta||\tilde{\pi}), D_{\text{KL}}^{d_{\tilde{\pi}}}(\tilde{\pi}||\pi_\theta))$ and $\delta_2 = D_{\text{KL}}^{d_\pi}(\tilde{\pi}||\pi)$, we have

$$\eta(\pi_\theta) - \eta(\pi) \geq -\frac{\sqrt{2}}{1-\gamma} \delta_1^{\frac{1}{2}} M^{\pi_\theta} + \frac{1}{(1-\gamma)\beta} \delta_2 - \frac{\sqrt{2}\gamma\epsilon_\pi^{\tilde{\pi}}}{(1-\gamma)^2} \delta_2^{\frac{1}{2}} \tag{26}$$

$\square$

# B  Discussion

## B.1  Connection with regularized policy optimization

In this subsection, we show in a general form, the proposed procedure of Algorithm 1 can recover many interesting algorithms, which are related to previous works. We consider a general regularized policy optimization problem (RPO) as

$$\max_{\theta \in \Theta} \left( (1-\gamma) L^{d_\pi,\pi}(\pi_\theta) - \frac{1}{\beta} D_{\text{app}}(\pi, \pi_\theta) \right) \tag{27}$$

where $D_{\text{app}}$ is a divergence use for approximation (e.g. KL divergence $D_{\text{KL}}$, Bregman divergence $D_\psi$). A closely related formulation of Algorithm 1 is

$$\min_{\theta \in \Theta} D_{\text{app}}(\mathcal{O}(\pi), \pi_\theta), \quad \text{where } \mathcal{O}(\pi) = \arg\max_{\pi' \in \Pi} \left( (1-\gamma) L^{d_\pi,\pi}(\pi') - \frac{1}{\beta} D_{\text{KL}}(\pi'||\pi) \right) \tag{28}$$

We call this generalized method as *imitating a better policy* (IBP).

### B.1.1  RPO, with $D_{\text{app}}(\pi, \pi_\theta) = D_{\text{KL}}^{d_\pi}(\pi||\pi_\theta)$

When $D_{\text{app}}(\pi, \pi_\theta)$ is realized with $D_{\text{KL}}^{d_\pi}(\pi, \pi_\theta)$, the RPO problem is equivalent to the constrained policy optimization problem considered in [Schulman et al., 2015]. The optimization objective is

$$\arg\max_{\theta \in \Theta} (1-\gamma)\beta L^{d_\pi,\pi}(\pi_\theta) - D_{\text{KL}}^{d_\pi}(\pi||\pi_\theta) = \arg\max_{\theta \in \Theta} \mathbb{E}_\pi \left( \frac{\pi_\theta(a|s)}{\pi(a|s)} \beta A^\pi(a|s) + \log \pi_\theta(a|s) \right) \tag{29}$$

### B.1.2  RPO, with $D_{\text{app}}(\pi, \pi_\theta) = D_{\text{KL}}^{d_\pi}(\pi_\theta||\pi)$

When $D_{\text{app}}(\pi, \pi_\theta)$ in RPO is set to the forward KL divergence $D_{\text{KL}}^{d_\pi}(\pi_\theta, \pi)$, the optimization objective becomes

$$\arg\max_{\theta \in \Theta} (1-\gamma)\beta L^{d_\pi,\pi}(\pi_\theta) - D_{\text{KL}}^{d_\pi}(\pi_\theta||\pi) = \arg\max_{\theta \in \Theta} \mathbb{E}_\pi \left( \frac{\pi_\theta(a|s)}{\pi(a|s)} \left( \beta A^\pi(a|s) - \log \frac{\pi_\theta(a|s)}{\pi(a|s)} \right) \right) \tag{30}$$

### B.1.3  IBP, with $D_{\text{app}}(\pi, \pi_\theta) = D_{\text{KL}}^{d}(\pi||\pi_\theta)$

For IBP as in (28), when $D_{\text{app}}(\pi, \pi_\theta) = D_{\text{KL}}^d(\pi||\pi_\theta)$, the optimization objective becomes

$$\arg\max_{\theta \in \Theta} -D_{\text{KL}}^d(\mathcal{O}(\pi)||\pi_\theta) = \arg\max_{\theta \in \Theta} \mathbb{E}_{s \sim d(s), a \sim \pi(a|s)} \exp(\beta A^\pi(s,a) + C(s)) \log(\pi_\theta(a|s)) \tag{31}$$

This is the main algorithm 12 discussed in our work.

Table 2: Connection between the proposed imitating a better policy (IBP) procedure and previous policy optimization methods

| Method | $D_{\text{app}} = D_{\text{KL}}(\pi \| \pi_\theta)$ | $D_{\text{app}} = D_{\text{KL}}(\pi_\theta \| \pi)$ |
|---|---|---|
| RPO | $\approx$ TRPO (29) | IBP with KL (30) |
| IBP (28) | MARWIL (31) | IBP with KL (30) |

### B.1.4 IBP, with $D_{\text{app}}(\pi, \pi_\theta) = D_{\text{KL}}^{d_\pi}(\pi_\theta \| \pi)$

When $D_{\text{app}}(\pi, \pi_\theta) = D_{\text{KL}}^{d_\pi}(\pi \| \pi_\theta)$, the algorithm IBP is equivalent to RPO as in Formula (30). In general, for a family of functions $\psi(\cdot)$ we define the Bregman divergence as $D_\psi^d(\pi', \pi) = \sum_s d(s) \Delta_\psi(\pi'(\cdot|s), \pi(\cdot|s))$, where $\Delta_\psi(x, y) = \psi(x) - \psi(y) - \langle \nabla \psi(y), x - y \rangle$, with the inner product $\langle \cdot, \cdot \rangle$ taken on the action space $\mathcal{A}$. We then have

$$\max_{\theta \in \Theta}((1-\gamma)\beta L^{d_\pi, \pi}(\pi_\theta) - D_\psi(\pi_\theta, \pi)) \tag{32}$$

equivalent to the problem

$$\min_{\theta \in \Theta} D_\psi(\pi_\theta, \mathcal{O}(\pi)), \quad \text{where } \mathcal{O}(\pi) = \arg \max_{\pi' \in \Pi}((1-\gamma)\beta L^{d_\pi, \pi}(\pi') - D_\psi(\pi', \pi)) \tag{33}$$

To summarize, the proposed procedure is closely related to existing methods like regularized policy optimization. And for special cases, imitating a better policy is equivalent to regularized policy optimization. We present the relationship with previous method in Table 2.

### B.2 Additional Details of the Algorithm Settings for HFO

To parametrize the policy and value function, we use a neural network with multiple outputs and shared basic layers. 3 fully connected layers are used as the shared base layers, each having 64 hidden nodes and followed by an ELU [Clevert et al., 2015] activation layer. For outputting probability for discrete actions $k = 1, 2, 3$, a small network of 2 fully connected layers with 32 hidden nodes and 3 soft-maxed outputting nodes are appended to the base layers. For outputting the mean of normal distribution for each action's parameter, we use a $32 \times 6$ fully connected 2 layer network after the base layers. We also use a third $32 \times 1$ network appended to the base layers to output the state value $V(s_t)$ for each state.

In our implementation, we distribute the algorithm over different servers to speed up the experiments. The "worker" processes which are responsible for generating trajectories to be filled in the replay memory $D$ are deployed on a CPU server. A "trainer" process which is responsible for updating $\theta$ is deployed on a GPU server. The replay memory is distributed over the cluster to collect trajectories from workers in parallel and provide batches of data for the trainer.

For these experiments, we use 20 workers and 1 centralized trainer. The maximum capacity of the replay memory for each actor is set to 32 episodes, meaning a total of 640 episodes. In each iteration we randomly sample a batch of 1024 samples from $D$. The overall loss is the policy loss plus the squared Bellman error of $V^\pi$. The basic learning rate is set to $10^{-4}$, with $\beta$ set to $1.0$. The learning rate decreases proportional to $1/\sqrt{0.0001T}$. We use RMSProp with weight decay set to $10^{-5}$ and no momentum. In the experiment, each run is allowed to iterate 100000 batches to converge.

### B.3 Additional Details of the Algorithm Settings for King of Glory

The solo mode of King of Glory is similar to those in previous works [Jiang et al., 2018, Xiong et al., 2018], except that we use the hero *Diao Chan* in our experiments. For quantitatively measure, we use a pool of AI agents as opponents, and calculate the Elo ratings [Coulom, 2005] of the agents trained with/without the proposed technique. Experimental results are summarized in Table 3. As can be seen, the agents trained with our proposed method are significantly stronger than those trained with the baseline method (IL). Also, in a proprietary test with colleagues, the *Diao Chan* AI can defeat experienced *XingYao* and *WangZhe* level [4] players in a solo game. We conclude that the proposed method can be successfully used in training AI agents for complex video games with hybrid action space in real-world.

**Feature** For each frame, we extract 4 types of feature to represent the game state:

1. **Image-like Feature of Global View** The image-like feature covers the whole map of solo mode, with a resolution of $16 \times 64$ and 6 channels of allied hero position, allied soldiers' positions, allied towers' defense region, enemy hero position, enemy soldiers' positions, and enemy towers' defense region.

Table 3: Performance of the AI agents trained with/without the proposed technique. A total of 40 AI agents trained with different methods are tested in roughly round-robin matches. For comparison, MARWIL1 and IL1 use the same state-features, network structure, and algorithm settings, except for the update formula when computing the gradient. Similarly, MARWIL2 and IL2 use the same settings except for the update formula. The Elo score (higher is better) measures the strength of agents, and the winning ratio is the percentage of games the agent has won. In the results, the best agent is trained with MARWIL method, which reaches an Elo score (higher is better) of 126, and a winning ratio of 64%.

| AGENT | ELO | W.RATIO |
|---|---|---|
| MARWIL1 | 126 | 64% |
| MARWIL2 | 72 | 58% |
| IL1 | -65 | 41% |
| IL2 | -184 | 26% |

2. **Image-like Feature for Local View** The image-like feature corresponds to the player's screen size of map. The resolution is $32 \times 48$ with 12 channels of allied hero position, allied hero attack region, allied soldiers' positions, allied soldiers' HP, allied towers' defense region, allied bullets' damage region, enemy hero position, enemy hero attack region, enemy soldiers' position, enemy soldiers' HP, enemy towers' defense region, and enemy bullets' damage region.

3. **Dense Feature** A 256 dimension dense feature is extracted for each frame. These features include allied and enemy heroes' basic attributes and properties, towers' status and soldiers' status, etc.

4. **Sparse Feature** Two sparse features are provided to indicate the allied and enemy hero types.

**Action** We use a hybrid of discrete and continuous action space. The action space is defined as $\mathcal{A} = K \times \mathbb{R}^2$, where $K = 6$. The 6 discrete action types are: `NoAction`, `Move`, `Attack`, `Skill1`, `Skill2`, `Skill3`. For $k \in \{$ `Move`, `Skill1`, `Skill2` $\}$, the environment also accepts a "direction" $x_k \in \mathbb{R}^2$ as the action parameter.

**Reward** We craft 14 dimension rewards as the optimization target, namely `ShortTimeGold`, `LongTimeGold`, `InstantHP`, `ShortTimeHP`, `Kill`, `Death`, `Exp`, `LevelUp`, `Damage`, `DamageToHero`, `TowerDestruct`, `HighTowerDestruct`, `CrystalDestruct`, and `WinLoss`. Different rewards $r^{(k)}$ may have different discount factors $\gamma^{(k)}$. A weighted sum of $R_t^{(k)}$ is used as the final cumulative reward $R_t = \sum_{k=0}^{13} w^{(k)} R_t^{(k)}$, where $R_t^{(k)} = \sum_{l=t}^{T} (\gamma^{(k)})^{l-t} r_l^{(k)}$.

**Network Structure** We adopt a VGG [Simonyan and Zisserman, 2014] like structure for image-like features: Each "block" consists of 5 layers in the order of Conv-ELU-Conv-ELU-Pooling, with kernel size of 3 for convolution and stride of 2 for max-pooling. By default we use ELU [Clevert et al., 2015] as activation layer for convolution layers and fully-connected layers. For global view image-like feature, 3 "blocks" of size $32 \times 16 \times 64$ [5] $\rightarrow 64 \times 8 \times 32 \rightarrow 64 \times 4 \times 16$ are stacked to extract information from raw image-like features. For local view image-like feature, 3 "blocks" of size $32 \times 16 \times 24 \rightarrow 64 \times 8 \times 12 \rightarrow 64 \times 4 \times 6$ are stacked. Each sparse feature is embedded to a vector of 32 dimension and concatenated together with the dense feature, followed by 2 fully connected layers with 512 hidden nodes. Then these preprocessed representations from image-like features and dense features are all concatenated, followed by 5 fully connected layers of 2048 hidden nodes. 3 final modules consisting of two fully-connected (FC) layers of hidden size 512 and 256 are appended for outputting discrete action probabilities, continuous action parameters, and value estimation for each dimension of rewards, respectively. The whole network structure is depicted as in Figure 2.

Figure 2: Network structure for our AI agent in King of Glory

## Footnotes

[4] The level of a player in the mobile game is ranked (from lowest to highest) by *QingTong*(Bronze), *BaiYin*(Silver), *HuangJin*(Gold), *BoJin*(Platium), *ZuanShi*(Diamond), *XingYao*(Starshine), and *WangZhe*(King).

[5]We write $C \times H \times W$ for brevity, where $C$ is the number of channels, $H$ is the height, and $W$ is the width.