[Reviews · NeurIPS 2018]

Reviewer 1



A method for learning deep policies from data recorded in demonstrations is introduced. The method uses exponentially weighted learning that can learn policies from data generated by another policy The proposed approach is interesting and well presented. it opens the possibilities for future work, more specifically to RL, as stated in the conclusions. Theat would be even more interesting than this presented imitation learning scheme, however the paper gives the introduction, background and discussion for that future work. How is generated the data for the HFO environment? Why is not used PG, PGIS in the experiments with Torcs and king of Glory? I suggest to specify in Table 1 that B=0.0 is the case of IL. Since the approach is considered Imitation Learning. I consider sensitivity analysis (like the carried out for HFO in Fig 1) of noisy data very important, because most of the time that is the case with the Demonstrations. I suggest to include it for the Torcs and king of Glory environments. I found very troublesome that in section 6.3 the Table 2 shows results of EWIL1, EWIL2, IL2, and IL2, however it is never mentioned what are the difference in those settings between the case 1 and 2, i.e. that does not contribute anything. I consider the section 6.3 can be eliminated since the results are not illustrating something richer than: EWIL is somehow better than IL. If it had comparisons with other methods and also varying parameters like section 6.1 that would be better. Similarly to the previous comment, experiments with Torcs could have shown more interesting observations, so those results are contributing very little. Additionally, it lacks of conclusions about the results, based only on that table, to me it seems that playing with that parameter is worthless since the best score is with the highest value of B=1. Does it always happen for all the learning problems?. Does it have any disadvantage to set B=1? Again more results and analysis could have been taken with those environments used in the experiments.

Reviewer 2



Summary: The paper considers imitation learning with only batched historical trajectories and without any further access to any simulators or environment models. The proposed imitation learning is reward-aware, in a sense that it weights the expert's action based on advantage of the expert policy A^{\pi}. The paper also analyze the proposed algorithm and show a policy improvement over the expert policy. Strengths: The setting considered in the paper is challenging: the learning algorithm only has access to batched trajectories from the expert but further access to the simulator or model. Hence most of previous imitation learning works (e.g., Dagger ,GAIL) won't apply in this setting. Simply Behavior cloning will work in this setting, but the paper shows that by leveraging the advantage of the expert, it can outperform Behavior cloning. Main comments below: 1. The definition and the use of KL divergence between two policies, i.e., D_{KL}(\pi'||pi) is not clear. Specifically, the definition in Eq (1) is different from the definition of KL below line 56. Which definition did you use in the rest of the paper, say, in Eq (3)? 2. Regarding the state distribution rho(s), I assume it is always referring to the state distribution of the behavior policy pi. However, below line 146, when you say that it is imitating a new hypothetic policy \tilde{\pi}, shouldn't the state distribution becomes the state-distribution resulting from \tilde{\pi}? The state distribution of tilde{\pi} and \pi could be different. 3. Line 147: Why C(s) could be emitted? It is state-dependent. Ignoring C(s) could affect the solution of the optimization. Could the authors elaborate on why it is ok to ignore C(s)? 4. Even assume that we have a rich enough policy space such that we can drive D_{KL}(\tilde{\pi} || \pi_{\theta}) to zero. How does this guarantee that \pi_{\theta} is also as good as, or even better than pi? Note that D_{KL}(\tilde{\pi}|| \pi) = 0 does not imply that \tilde{\pi} and \pi are the same under every state: it depends on whether or not state distribution rho(s) has non-zero probability on every state (consider the case where. a behavior policy can only cover a small part of the large state space). Also no matter how deeper the network is, we can never ever make D_{KL}(\tilde{\pi}|| \pi_{\theta}) reach zero, simply because we do not have infinitely many data. Assuming, for instance, D_{KL}(\tilde{\pi} || pi_{\theta}) <= epsilon for some small epsilon, is reasonable, but then we need to analyze how epsilon affect the performance of the learned \pi_{\theta}. 5. Notations in Eq (7) - (10) are not well-defined. Could you point the readers to the places where, for instance, E_{\pi} \log \pi_{\theta} is defined? What does it even mean by log(\pi_{\theta}), as pi_{\theta} is a function? Also the KL used in (7) and (10) are not consistent. Following the definition in (7), shouldn't the expectation in (10) be defined with respect to \tilde{\pi}? After rebuttal: Thanks a lot for the rebuttal. I read all reviews and the rebuttal. Unfortunately I'm not going to raise my score. Regarding the theory section, essentially the paper is trying to imitate an ideal policy (\tilde{\pi} in Eq 4) using function approximations (\pi_{\theta}). The paper claims the algorithm 1 essentially is minimizing D_{KL}(\tilde{\pi} || \pi_{theta}), but I'm not sure if it's a good idea. Again, the state distribution problem plays important role here. Note that in Alg 1, it is using states from the behavior policy, which is neither \tilde{\pi} nor \pi_{\theta}!. (Existing analysis of Behavior cloning assumes we generate state-action pairs from the policy that we are going to imitate! However here we do not have data from \tilde{\pi}---the policy that we are supposed to imitate). To behavior clone this ideal policy, one should minimize KL under the state-distribution resulting from the ideal policy \tilde{\pi}. This is extremely important for the discussion in section 5.2. While it's not hard to see \tilde{\pi} is a uniformly better policy than the behavior policy \pi, it is definitely not clear to me that the learned policy pi_{\theta} would be better than or at least as good as \pi, considering that the state-distribution for measuring KL-divergence between \tilde{\pi} and \pi_{\theta} is the state-distribution of the behavior policy \pi. Overall, I might be wrong but I personally think that the paper somehow implicitly kept using max_{s} KL(\pi(. | s) || \pi'(. | s) ) as the default-to-go definition of KL-divergence between two policies, at least throughout the analysis (if we can drive this kl-divergence to zero or small number epsilon, then pi and pi' will be close to each other for any states, hence in analysis we do not need to worry about the state-action distribution). But in reality, we can only measure KL under the state-distribution of behavior policy (i.e., max_{s} is impossible to evaluate in large state space). Theoretical guarantees of imitation learning where state-action distribution induced neither from the expert policy, i.e., \tilde{\pi} (classic Behavior cloning analysis considers this setting), nor from the learned policy, i.e. \pi_{\theta} (method like DAgger considers this setting) seem not trivial and need more careful analysis.

Reviewer 3



The paper tackles an important problem of learning a policy from historical data. In its setting, the data may be generated by an unknown policy, and have complex action space, thus making existing methods hard to apply. The authors propose a novel method which successfully solves these problems. The effectiveness is validated with theoretical justifications and experimental results. In a nutshell, the authors provide an elegant solution for the problem considered, where previous methods are likely to fail. Related works are discussed in Section 3, with the difference from their work, as well as the reasons the existing methods cannot be directly applied in their problem. From the analysis and the experiments, I think the proposed method is simple and welly suited for their problems of deep learning in complex games. Clarity: The writing is mostly clear and rigorous. Most notations and symbols used in this paper are defined in Section 2. Personally, I think formula (7) - (10) in Section 6 can be written more strictly without omitting s and a, although I get what the authors mean. Significance: I think the authors tackled an interesting and important problem in practice. I like the solution the authors proposed, which is both simple and effective. From the results in their experiments, the improvement over baseline methods is significant. In my opinion the work is likely to inspire a few future works. Questions: In Figure 1, I understand that a fair comparison should be made between EWIL and IL, which clearly shows EWIL a winner. However I am a little troubled by the gap between IL and the behavior policy. Do you have any analysis about why the performance of the cloned policy is better than the behavior policy? Since in IL you do not use any information from reward r, I expect their performance should be comparable. I guess this gap is related to the function approximation adopted? Overall, I think the paper tackles an important and interesting problem in a creative and effective way. The proposed method is analytic justifiable and empirical validated. The writing is clear and correct. For the above reasons, I think the paper ought to be accepted. ######################After Rebuttal ################################ After reading the feedback from the authors and the comments from other reviewers, I want to append some extra opinions in my comments. 1) I agree with the authors that the state distribution does not change the optimal discriminative classifier, which is the optimal policy conditioned on each state. 2) In Sec 5.1, the authors show the connection of Alg 1 and imitating a better policy. The approximation error may not be averaged on the state distribution of the improved policy. However the equivalence still holds with their definition of KL divergence with specific state distribution \rho(s). 3) For the function approximation part, personally, I think it is not important whether a policy is better than another in some states with zero state distribution probability. To me, it is a common assumption that every state considered in the state space will be visited many times, as in an ergodic MDP with infinite horizon.